# Statistically Profiling Biases in Natural Language Reasoning Datasets and Models

**Shanshan Huang**
Shanghai Jiao Tong University
Shanghai, China
`huangss_33@sjtu.edu.cn`

**Kenny Q. Zhu**[*]
University of Texas at Arlington
Arlington, Texas, USA
`kenny.zhu@uta.edu`

## Abstract

Recent studies have shown that many natural language understanding and reasoning datasets contain statistical cues that can be exploited by NLP models, resulting in an overestimation of their capabilities. Existing methods, such as "hypothesis-only" tests and CheckList, are limited in identifying these cues and evaluating model weaknesses. We introduce ICQ (I-See-Cue), a lightweight, general statistical profiling framework that automatically identifies potential biases in multiple-choice NLU datasets without requiring additional test cases. ICQ assesses the extent to which models exploit these biases through black-box testing, addressing the limitations of current methods. In this work, we conduct a comprehensive evaluation of statistical biases in 10 popular NLU datasets and 4 models, confirming prior findings, revealing new insights, and offering an online demonstration system to encourage users to assess their own datasets and models. Furthermore, we present a case study on investigating ChatGPT's bias, providing valuable recommendations for practical applications.

## 1 Introduction

Deep neural models have made remarkable strides in a broad spectrum of natural language understanding (NLU) tasks (Bowman et al., 2015; Wang et al., 2018; Mostafazadeh et al., 2016; Roemmele et al., 2011; Zellers et al., 2018). These tasks often employ a multiple-choice framework, as illustrated in Example 1. However, the inherent sensitivity of these models to minute variations calls for a robust and precise evaluation mechanism (Jurafsky et al., 2020).

**Example 1** *Natural language inference in the SNLI dataset, with the correct answer bolded.*

*Premise: A swimmer playing in the surf watches a low flying airplane headed inland.*

*Hypothesis: Someone is swimming in the sea.*

*Label: **a) Entailment.** b) Contradiction. c) Neutral.*

In tasks akin to Example 1, humans typically rely on the logical relationship between the premise and hypothesis. Contrarily, some NLP models might bypass this logical reasoning, focusing instead on the biases embedded within the dataset and, more specifically, within the hypotheses (Naik et al., 2018; Schuster et al., 2019). These biases—such as sentiment or shallow n-grams—could provide misleading cues for correct predictions.

We refer to these biases as "artificial spurious cues" when they pervade both the training and test datasets, maintaining a similar distribution over predictions. An example of such a cue is a model's disproportionate dependence on the word "someone" in Example 1. These cues, when absent or altered, can significantly impair a model's performance, underlining the importance of identifying them to enhance model robustness in the future.

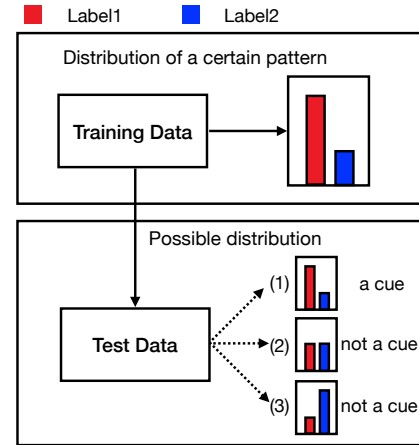

Figure 1: Example of a *cue*.

To tackle the issue of cues, it's crucial to distinguish between cues embedded in the dataset and those learned by the model. Conventional bias

---

[*]The corresponding author.

detection and mitigation tools, such as the AI Fairness 360 toolkit (Bellamy et al., 2018), primarily target dataset biases, inadequately addressing those learned by models during training.

While existing methods like "hypothesis-only" tests and CheckList can uncover model vulnerabilities, they're not expressly designed to identify model-learned cues. "Hypothesis-only" tests can highlight dataset issues where the hypothesis alone can provide a correct answer but fail to realistically depict the model's capabilities as they don't evaluate the model using the full data context used during both training and prediction.

Drawing from the tenets of black-box testing in software engineering, CheckList scrutinizes model weaknesses without detailed knowledge of the model's internal architecture. It achieves this by delivering additional stress test cases premised on predefined linguistic features. However, CheckList's dependence on meticulously designed templates limits its scope, and it also falls short of illuminating the knowledge the model has actually gleaned from the data.

To address these limitations, we introduce ICQ ("I-see-cue"), a resilient statistical analysis framework [1] engineered to identify model-learned cues. Diverging from traditional methods, ICQ identifies biases in multiple-choice NLU datasets without necessitating additional test cases. Employing black-box testing, ICQ assesses how models utilize these biases, delivering a comprehensive understanding of the bias in NLU tasks.

We authenticate ICQ's efficacy by deploying it on various NLU datasets to probe potential cues learned by models during training. ICQ facilitates an in-depth understanding of how models like Chat-GPT [2] learn potential cues, and it offers illustrative examples to guide the selection of suitable prompts, providing invaluable guidance for model optimization.

In summary, this paper contributes the following:

- We unveil ICQ, a lightweight yet potent method for identifying statistical biases and cues in NLU datasets, proposing simple and efficient tests to quantitatively and visually evaluate whether a model leverages spurious cues in its predictions.

---

[1]The code and dataset are available at https://github.com/flora336/icq
[2]https://chat.openai.com/

- We execute a comprehensive evaluation of statistical bias issues across ten popular NLU datasets and four models, corroborating previous findings and unveiling new insights. We also offer an online demonstration system to showcase the results and invite users to evaluate their own datasets and models.

- Through a case study, we delve into how Chat-GPT learns potential biases, offering valuable recommendations for its practical applications.

## 2 Preliminary

### 2.1 Task Definition

We define an instance $x$ of an NLU task dataset $X$ as

$$x = (p, h, l) \in X, \qquad (1)$$

where $p$ is the context against which to do the reasoning ($p$ corresponds to "premise" in Example 1); $h$ is the hypothesis given the context $p$; $l \in \mathcal{L}$ is the label that depicts the type of relation between $p$ and $h$. The size of the relation set $\mathcal{L}$ varies with tasks.

### 2.2 Linguistic Features

As demonstrated in previous work (Naik et al., 2018; Jurafsky et al., 2020), we consider the following linguistic features:

**Word**: The existence of a specific word in the premise or hypothesis of a dataset instance.

**Sentiment**: The sentiment value of an instance, calculated as the sum of sentiment polarities of individual words.

**Tense**: The tense feature (past, present, or future) of an instance, determined by the POS tag of the root verb.

**Negation**: The existence of negative words (e.g.,"no", "not", or"never") in an instance, determined by dependency parsing.

**Overlap**: The existence of at least one word (excluding stop words) that occurs in both the premise and hypothesis.

**NER**: The presence of named entities (e.g., PER, ORG, LOC, TIME, or CARDINAL) in an instance, detected using the NLTK NER toolkit.

**Typos**: The presence of at least one typo in an instance, identified using a pretrained spelling model.

For multiple-choice datasets, all features except Overlap are applied exclusively to hypotheses.

## 3 Approach

The ICQ framework, depicted in Figure 2, consists of three phases: data extraction, cue discovery, and model probing. In the data extraction phase, instances containing a specific linguistic feature $f$ are extracted from the dataset. The cue discovery phase identifies potential cues among pre-defined features. Finally, the model probing phase conducts two tests: the "accuracy test" and "distribution test". We will discuss these phases in more detail below.

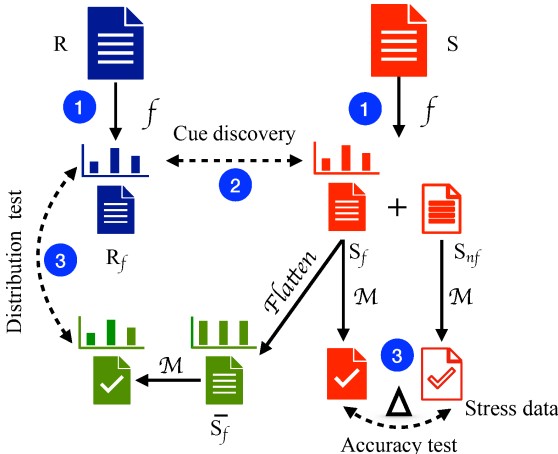

Figure 2: ICQ Workflow. ①: data extraction phase; ②: cue discovery phase; ③: model probing phase. $f$=a specific feature, $R$=training data, $S$=test data, $R_f$=extracted training data, $S_f$=extracted test data, $S_{nf}$=remaining test data without feature $f$, $\overline{S_f}$=flatten test data, $\mathcal{M}$=a specific model.

### 3.1 Data Extraction Phase

After defining the linguistic features, our system's fundamental step is constructing a data extractor for each feature value $f$. An extractor processes a dataset and retrieves a set of instances associated with the specified feature value. More specifically, for a specific feature, if an instance contains that feature, then that instance will be singled out as a part of the subset bearing that feature.

### 3.2 Cue Discovery Phase

For each feature $f$, we apply its extractor to both the training and test data of dataset $X$, denoted as $R$ and $S$ in Figure 2. This results in clustered subsets of training instances ($R_f$) and test instances ($S_f$). A feature is considered a possible cue for a dataset only if it is present in both the training and test data.

The bias of the label distribution for an extracted set is computed using the mean squared error

(MSE) and Jensen-Shannon Divergence (JSD) (Lin, 1991). The cueness score indicates the extent to which dataset $X$ is biased against a feature $f$.

$$MSE(F) = \frac{1}{|\mathcal{L}|} \sum_i (y_i - \overline{y_i})^2 \qquad (2)$$

Here, $y_i$ represents the number of instances with label $l_i$ in the extracted dataset $F$, and $\overline{y_i}$ is the mean number of instances for each label. A larger $MSE(F)$ implies a more pointed label distribution and greater bias. If the extracted training set ($R_f$) and the extracted test set ($S_f$) exhibit similar biases, the JSD between their distributions will be small:

$$JSD = \frac{1}{2}\left(Q(R_f) \parallel A\right) + \frac{1}{2}\left(Q(S_f) \parallel A\right), \quad (3)$$

where $A = \frac{1}{2}\left(Q(R_f) + Q(S_f)\right)$. The function $Q()$ denotes the label distribution of the extracted dataset. We define the cueness score as:

$$cue(f, X) = \frac{MSE(R_f)}{\exp(JSD(R_f, S_f))} \qquad (4)$$

This cueness score represents the degree to which a dataset $X$ is biased against a feature $f$.

### 3.3 Model Probing Phase

In the previous section, we established that a dataset $X$ could be influenced by a cue $f$. However, a model trained on this dataset may not necessarily exploit that cue, as models depend on both data and architecture. In this section, we introduce a framework to probe any model instance trained on the biased dataset, assessing if it utilizes cue $f$ and to what degree. We achieve this through two tests: the *accuracy test* and the *distribution test*.

#### 3.3.1 Accuracy Test

The accuracy test examines the model's performance on data subsets with and without specific features. By comparing the model's accuracy on these two subsets, we can understand the model's generalization capabilities under different conditions. If the model's performance shows a noticeable improvement on the subset containing a specific feature, it may suggest that the model has exploited that feature for prediction.

In the accuracy test, we assess the prediction accuracies of the model $M$ on the extracted test set (with feature $f$) and on the remaining test set (without feature $f$), denoted as $acc(S_f)$ and $acc(S_{nf})$,

respectively. The accuracy test computes the difference between these two accuracies:

$$\Delta Acc(f) = acc(S_f) - acc(S_{nf}) \quad (5)$$

A positive or negative value of $\Delta Acc$ indicates the direction of the model's performance change when comparing its accuracy on data subsets with and without specific features. A positive value suggests that the model exploits the feature for prediction, while a negative value implies struggles to generalize or detrimental sensitivity to the feature.

The magnitude of the absolute value of $\Delta Acc$ reflects the degree to which a model's performance is affected by the presence or absence of specific features in the data subsets. A larger absolute value indicates a stronger reliance on or sensitivity to the feature, whereas a smaller absolute value suggests a more robust model that is less affected by the presence or absence of the feature.

### 3.3.2 Distribution Test

The distribution test is a visual examination that focuses on how changes in specific feature distributions within datasets affect a model's predictive performance.

First, we create a "stress dataset" $\overline{S_f}$ by "flattening" the label distribution in $S_f$. We achieve this by removing random instances from all labels except for the one with the smallest number of instances, stopping when a balance is reached. In other words, we retain the minimum number of instances present in each label, while randomly discarding the excess instances. This approach effectively eliminates bias in the extracted test set, and challenges the model.

Next, we apply the model to the stress test set and obtain prediction results. We then compare the label distribution of the prediction results on the stress test set with the label distribution of the extracted training data ($R_f$). The rationale is that if the extracted training data contains a cue, its label distribution will be skewed towards a specific label. If the model exploits this cue, it will prefer to predict that label as much as possible, amplifying the skewness of the distribution, despite the input test set being neutralized. We aim to observe such amplification in the output distribution to identify the model's weaknesses.

In summary, the accuracy test and distribution test are related in terms of assessing a model's sensitivity to specific features, but they emphasize different aspects. Distribution testing focuses on the impact of feature distribution changes on model performance, while accuracy testing evaluates the model's performance on data subsets with and without specific features. By combining these two testing methods, a model's sensitivity to particular features can be more comprehensively assessed. If both tests determine that the model is sensitive to a certain feature, we can have a higher degree of confidence in this conclusion.

## 4 Evaluation

We first present the experimental setup, followed by results on cue discovery, model probing, and analysis. The entire framework is implemented in an online demo.

### 4.1 Setup

We evaluate this framework on 10 popular NLR datasets in Table 1 and 4 well-known models, namely FASTText (FT) (Joulin et al., 2017), ESIM (ES) (Chen et al., 2016), BERT (BT) (Devlin et al., 2018) and RoBERTA (RB) (Liu et al., 2019) on these datasets. All these datasets except for SWAG (Zellers et al., 2018) and RECLOR (Yu et al., 2020) are collected through crowdsourcing. SWAG is generated from an LSTM-based language model. Specifications of the datasets are listed in Table 1.

| Dataset | Type | Data Size | Train/Test | Human Acc |
|---|---|---|---|---|
| | | | Ratio | (%) |
| SNLI | CLS | 570K | 56:1 | 80.0 |
| QNLI | CLS | 11k | 19:1 | 80.0 |
| MNLI | CLS | 413k | 40:1 | 80.0 |
| ROC | MCQ | 3.9k | 1:1 | 100.0 |
| COPA | MCQ | 1k | 1:1 | 100.0 |
| SWAG | MCQ | 113k | 4:1 | 88.0 |
| RACE | MCQ | 100k | 18:1 | 94.5 |
| RECLOR | MCQ | 6k | 9:1 | 63.0 |
| ARCT | MCQ | 2k | 3:1 | 79.8 |
| ARCT_adv | MCQ | 4k | 3:1 | - |

Table 1: 10 Datasets. Data size refers to the number of questions in each dataset. CLS=Classification. MCQ=Multiple Choice Questions. By our definition, $k$-way MCQs will be split into $k$ instances in preprocessing.

These datasets can mainly be classified into two types of tasks. SNLI, QNLI, and MNLI (Williams et al., 2018) are classification tasks, while ROC, COPA (Roemmele et al., 2011), SWAG, RACE (Lai et al., 2017), RECLOR, ARCT (Habernal et al., 2017) and ARCT_adv (Schuster et al., 2019) are multiple-choice reasoning tasks. Features appearing a minimum of five times in training

or testing sets are considered cues.

## 4.2 Cues in Datasets

In this section, we showcase the cues identified in each dataset using the cueness metric, as described in Section 3.

We first filter the training and test data for each dataset using all the features defined in this paper. The left half of Table 2 displays the top 5 cues discovered for each of the 10 datasets, along with their cueness scores. ARCT_adv, an adversarial dataset, is intentionally well-balanced. Consequently, we only found one cue, OVERLAP, with a very low cueness score. This is unsurprising since OVERLAP is the only "second-order" feature in our list of linguistic features that considers tokens in both the premise and hypothesis, and likely evaded data manipulation by the creator.

Mostly, the top 5 cues discovered are word features. However, besides OVERLAP, we also see NEGATION and TYPO appearing in the lists. In fact, SENTIMENT and NER features would have emerged if we expanded the list to the top 10. Interestingly, several features previously reported as biased by other works, such as "not" and NEGATION in ARCT, "no" in MNLI and SNLI, and "like" in ROC, are also found. Particularly in MNLI, all five discovered cues are related to negatively toned words, suggesting significant human artifacts in this dataset that can lead to model fragility.

Additionally, we observe that some word cues are indicative of certain syntactic, semantic, or sentiment patterns in the questions. For example, "because" in SNLI implies a cause-effect structure; "like" in ROC indicates positive sentiment; "probably" and "may" in RACE suggest uncertainty, and so on. These features can serve as clues for revising datasets.

## 4.3 Biases in Models

To investigate whether a model is affected by a specific cue or feature in a dataset, we train four models on their original training sets and evaluate them using accuracy and distribution tests.

**Accuracy Test:** The results are presented in Table 2. As mentioned in Section 3.3.1, a positive or negative $\Delta$ value indicates the direction of a model's performance change when comparing its accuracy on data subsets with and without specific features. The absolute value of $\Delta$ reflects the degree to which the model's performance is influenced by these features, with larger values suggesting stronger reliance or sensitivity and smaller values indicating a more robust model.

The bottom of Table 2 shows that, across all 10 datasets, the sum of the absolute values of $\Delta$ follows the order: RoBERTA < BERT < ESIM < FastText. This is consistent with earlier hypothesis-only tests and the community's common perception of these popular models. However, examining individual datasets and features reveals a more nuanced situation. For instance, FastText tends to pick up individual word cues rather than semantic cues, while more complex models such as BERT and RoBERTA appear more sensitive to structural features like NEGATION and SENTIMENT, which are actually classes of words. This is well explained by FastText's design, which focuses more on modeling words than syntactic or semantic structures.

Interestingly, FastText exhibits a strong negative correlation with TYPO. We speculate that FastText might have been trained with a more orthodox vocabulary, making it less tolerant of typos in the text.

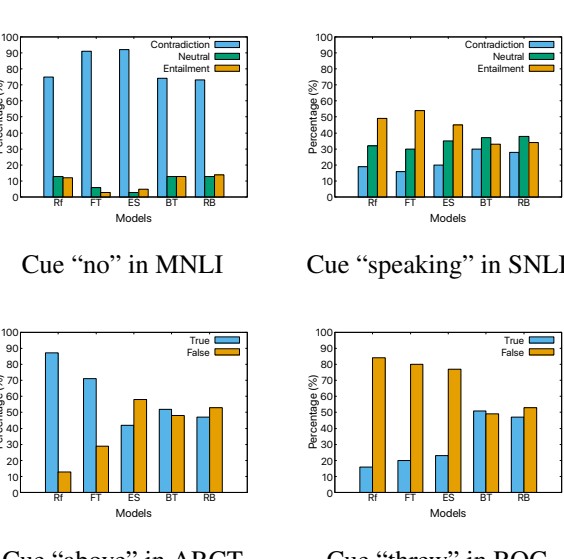

Cue "no" in MNLI     Cue "speaking" in SNLI

Cue "above" in ARCT     Cue "threw" in ROC

Figure 3: Four test examples for distribution comparison with 4 different models

**Distribution Test:** We highlight three interesting findings in Figure 3. The bars for the four models represent the distribution percentage based on each predicted label. $R_f$ denotes the extracted training data distribution with a specific feature. We observe that all models on the cue "no" in MNLI achieve positive $\Delta$ in Table 2, particularly FastText. Consistent with the "Accuracy Test," we find that the prediction label distribution skewness

| Dataset | Top Cues | Cueness % | FT (Δ) | ES (Δ) | BT (Δ) | RB (Δ) |
|---|---|---|---|---|---|---|
| SNLI | "sleeping" | 13.95 | 30.3 | 6.81 | 5.34 | 4.87 |
| | "no" | 13.33 | 18.09 | 3.32 | 2.05 | 2.6 |
| | "because" | 9.24 | 18.89 | 4.88 | 5.61 | 4.31 |
| | "friend" | 8.82 | 22.96 | 6.66 | 3.51 | 3.05 |
| | "movie" | 7.73 | 16.64 | 0.06 | 9.47 | -0.19 |
| QNLI | "dioxide" | 4.52 | 9.78 | -0.06 | 4.97 | 10.56 |
| | "denver" | 4.26 | 13.59 | 7.14 | 2.23 | 3.11 |
| | "kilometre" | 4.24 | 4.85 | 6.43 | 4.67 | 2.55 |
| | "mile" | 3.95 | 7.16 | 15.64 | -1.65 | -6.65 |
| | "newcastle" | 3.8 | 3.44 | 12.0 | 0.89 | -1.23 |
| MNLI | "never" | 10.4 | 29.15 | 26.41 | 9.86 | 10.6 |
| | "no" | 8.98 | 19.49 | 20.17 | 1.2 | 3.32 |
| | "nothing" | 8.98 | 25.5 | 26.84 | 5.11 | 4.32 |
| | "any" | 6.79 | 20.4 | 19.39 | 7.76 | 3.74 |
| | "anything" | 5.73 | 18.43 | 15.74 | 3.31 | 1.14 |
| ROC | "threw" | 12.99 | 1.28 | 4.69 | 10.88 | 0.97 |
| | "now" | 8.68 | -10.01 | 14.51 | 1.75 | 5.69 |
| | "found" | 8.16 | -2.31 | 4.45 | 5.12 | -3.13 |
| | "won" | 7.71 | 2.43 | 0.74 | 1.05 | 5.51 |
| | "like" | 7.3 | 4.77 | 10.06 | 8.81 | 1.67 |
| COPA | "went" | 3.61 | -10.83 | 6.46 | 7.92 | 1.04 |
| | "got" | 2.74 | 5.45 | -9.89 | -12.52 | -10.3 |
| | "for" | 2.14 | 10.11 | -1.89 | 9.05 | 11.58 |
| | "with" | 1.38 | -15.64 | -6.98 | 3.3 | 13.82 |
| | TYPO | 0.84 | -12.46 | -2.33 | 3.8 | -8.22 |
| SWAG | "football" | 7.38 | 6.13 | 8.55 | 1.2 | 1.55 |
| | "anxious" | 6.65 | 7.55 | -4.67 | -6.66 | -1.67 |
| | "concerned" | 6.19 | 12.6 | 4.58 | 8.27 | -5.66 |
| | "skull" | 5.73 | -2.77 | 0.49 | 8.43 | 3.49 |
| | "cop" | 5.01 | 2.79 | 5.3 | -0.92 | -0.04 |
| RACE | "above" | 13.74 | 8.73 | -8.43 | -0.22 | -1.92 |
| | "b" | 12.84 | 16.97 | -4.8 | 3.52 | -3.45 |
| | "c" | 11.83 | 15.69 | -6.94 | 8.6 | -7.6 |
| | "probably" | 6.77 | 9.91 | -0.06 | -3.8 | 2.86 |
| | "may" | 4.2 | 7.75 | -3.45 | -6.67 | -1.8 |
| RECLOR | "over" | 2.07 | 1.76 | -2.94 | -1.35 | -4.12 |
| | "result" | 1.97 | -3.29 | -2.69 | -1.78 | -3.7 |
| | "explanation" | 1.81 | -6.33 | -1.73 | -2.76 | -7.24 |
| | "proportion" | 1.68 | -5.64 | -4.69 | 2.37 | -2.16 |
| | "produce" | 1.4 | 4.54 | -2.98 | -14.36 | -3.7 |
| ARCT | "not" | 3.74 | -2.54 | 7.45 | -0.97 | -11.96 |
| | NEGATION | 2.85 | 3.49 | 10.04 | 6.28 | -8.23 |
| | "n't" | 2.52 | 10.3 | 5.89 | 9.49 | 4.84 |
| | "always" | 2.25 | -4.66 | 38.21 | -4.35 | -8.26 |
| | "doe" | 2.06 | -0.73 | -3.69 | -1.15 | -7.22 |
| ARCT_adv | OVERLAP | 1.96e-10 | 1.65 | -0.25 | 2.73 | 0.57 |
| $\sum(|.|)$ (Model weakness) | | | 469.8 | 361.4 | 227.7 | 216.2 |

Table 2: Datasets, their top 5 cues and 4 models biases $\Delta$ on them.

is amplified in Figure 3 for FastText and ESIM. With the "no" cue, they prefer to predict "Contradiction" even more than the ground truth in the training data. In contrast, BERT and RoBERTA moderately follow the training data. While the cue "no" is effective at influencing the models, the cue "above" is not as successful. Figure 3 shows that the distribution of predicted results for ESIM in ARCT is entirely opposite to the training data, explaining the $\Delta = -8.43$ in Table 2 and demonstrating that models may not exploit a cue even if it is present in the data. Similarly, "speaking" in BERT and RoBERTA can also explain their low $\Delta$ values, which are not shown in Table 2.

The example of the cue "threw" presents an out-

lier for BERT, as the distribution test result is inconsistent with the accuracy test: the accuracy deviation is very high for BERT, but its prediction distribution is flat. We have not encountered many such contradictory cases. However, when they do occur, as in this example, we give BERT the benefit of the doubt that it might not have exploited the cue "threw".

## 5 Case Study

Recently, ChatGPT, a large language model (LLM) released by OpenAI, has garnered significant interest from the NLP community. ChatGPT, a GPT-

**Prompt 1**

Analyze the logical relationship between the following
**premise**: '{premise}' and the **hypothesis**: '{hypothesis}'.
Determine whether the hypothesis is an *entailment*, *contradiction*, or *neutral* with respect to the premise.

**Prompt 1 + CoT**

Analyze the logical relationship between the following
**premise**: '{premise}' and the **hypothesis**: '{hypothesis}'.
Determine whether the hypothesis is an *entailment*, *contradiction*, or *neutral* with respect to the premise.
*Let's think step by step.*

**Prompt 2**

Please identify whether the premise entails, contradicts, or is neutral with respect to the hypothesis. The answer should be exactly "*entailment*," "*contradiction*," or "*neutral*."
**premise**: '{premise}'
**hypothesis**: '{hypothesis}'

**Prompt 2 + CoT**

Please identify whether the premise entails, contradicts, or is neutral with respect to the hypothesis. The answer should be exactly "*entailment*," "*contradiction*," or "*neutral*."
**premise**: '{premise}'
**hypothesis**: '{hypothesis}'
*, Let's think step by step.*

Figure 4: Prompts.

x [3] series model, is trained through Reinforcement Learning from Human Feedback (RLHF) (Christiano et al., 2017) similarly to InstructGPT (Ouyang et al., 2022). In this section, we investigate if zero-shot ChatGPT is influenced by bias features, using a case study focused on the word "no" in the MNLI dataset. We aim to compare the effectiveness of different prompts and select the best one for mitigating bias based on a single bias feature.

## 5.1 Dataset

We selected test instances from the MNLI dataset to study the influence of the word "no" on ChatGPT's performance. The original test set has *Contradiction*: 3240, *Entailment*: 3463, and *Neutral*: 3129 instances. Instances containing "no" are distributed as follows: *Contradiction*:229, *Entailment*: 38, and *Neutral*: 46.

For the accuracy test, we used all 313 instances with "no" and an equal number of instances without "no", randomly chosen from the remaining test set. This ensures a balanced evaluation of ChatGPT's performance.

For the distribution test, we selected 38 instances per label containing "no", resulting in a total of 114 instances.

## 5.2 Prompts

We have four prompts in Figure 4: The first prompt is proposed by ChatGPT itself. We ask, "What is the best prompt for the MNLI task according to you?". ChatGPT returns prompt 1 for us. The sec-

ond prompt is inspired by previous work (Qin et al., 2023). The third and fourth prompts are created by adding "Let's think step by step" (Kojima et al.) to prompt 1 and prompt 2 with the "chain of thought" (CoT) thinking, respectively. This modification has been shown to significantly improve the performance of InstructGPT on reasoning tasks (Ouyang et al., 2022).

## 5.3 ICQ Results

| Prompt | Acc ("no") | Acc (w/o "no") | $\Delta Acc$ |
|---|---|---|---|
| P1 | 74.34 | 77.32 | -2.98 |
| P2 | 75.42 | 74.18 | 1.24 |
| P1 + CoT | 78.35 | 77.28 | 1.07 |
| P2 + CoT | 76.67 | 76.40 | 0.27 |

Table 3: Accuracy test results (%). P1=Prompt 1, P2=Prompt 2.

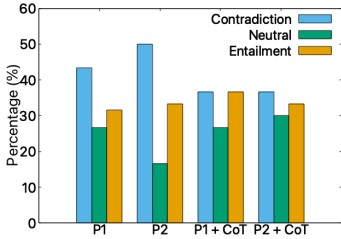

Figure 5: Distribution test results.

We evaluated the model's accuracy using different prompts on instances with and without the word "no." The results are shown in Table 3. P1 demonstrates a negative Δ Acc, indicating difficulty in generalizing when "no" is present. P2 exhibits a positive Δ Acc, suggesting better generalization.

---

[3] Currently, x is either 3.5 or 4, and in the subsequent experimental process, the ChatGPT we use is based on GPT-3.5.

Adding "CoT" to both prompts reduces bias risk. P1 + CoT shows the most significant improvement in Acc ("no"), but P2 + CoT has the smallest absolute Δ Acc, indicating the least sensitivity to "no" and the lowest bias risk among the tested prompts.

Besides, we analyzed the model's prediction distribution Figure 5 for the different prompts on the stress test set containing the word "no" with balanced label distribution. Distribution test results reveal imbalances in P1 and P2 prediction distributions, with P1 and P2 leaning towards predicting contradictions. Adding "CoT" mitigates these imbalances, leading to more balanced distributions. P2 + CoT presents the most balanced distribution among the labels, supporting its lowest bias risk.

In conclusion, our case study, particularly when focusing on the feature "no", indicates that zero-shot ChatGPT can be influenced by bias features. The choice of prompt significantly affects its performance. When assessing the feature "no", P2 + CoT showed the lowest bias risk across both tests. The "CoT" strategy appears effective in reducing bias risk for this specific feature. However, it's important to note that our findings, particularly regarding the feature "no", suggest that ChatGPT's self-recommended prompt (P1) might not always be optimal. This underscores the importance of human intervention and ongoing exploration to optimize performance and minimize bias risks. Future studies and conclusions would benefit from a more nuanced and feature-specific analysis.

## 6 Related Work

Our work is related to three research directions: spurious features analysis, bias calculation, and dataset filtering.

**Spurious features analysis** has been increasingly studied recently. Much work (Sharma et al., 2018; Srinivasan et al., 2018; Zellers et al., 2018) has observed that some NLP models can surprisingly get good results on natural language understanding questions in MCQ form without even looking at the stems of the questions. Such tests are called "hypothesis-only" tests in some works. Further, some research (Sanchez et al., 2018) discovered that these models suffer from insensitivity to certain small but semantically significant alterations in the hypotheses, leading to speculations that the hypothesis-only performance is due to simple statistical correlations between words in the hypothesis and the labels. Spurious features can be

classified into lexicalized and unlexicalized (Bowman et al., 2015): lexicalized features mainly contain indicators of n-gram tokens and cross-ngram tokens, while unlexicalized features involve word overlap, sentence length, and BLEU score between the premise and the hypothesis. (Naik et al., 2018) refined the lexicalized classification to Negation, Numerical Reasoning, Spelling Error. (McCoy et al., 2019) refined the word overlap features to Lexical overlap, Subsequence, and Constituent which also considers the syntactical structure overlap. (Sanchez et al., 2018) provided unseen tokens an extra lexicalized feature.

**Bias calculation** is concerned with methods to quantify the severity of the cues. Some work (Clark et al., 2019; He et al., 2019; Yaghoobzadeh et al., 2019) attempted to encode the cue feature implicitly by hypothesis-only training or by extracting features associated with a certain label from the embeddings. Other methods compute the bias by statistical metrics. For example, (Yu et al., 2020) used the probability of seeing a word conditioned on a specific label to rank the words by their biasness. LMI (Schuster et al., 2019) was also used to evaluate cues and re-weight in some models. However, these works did not give the reason to use these metrics, one way or the other. Separately, (Ribeiro et al., 2020) gave a test data augmentation method, without assessing the degree of bias.

**Dataset filtering** is one way of achieving higher quality in datasets by reducing artifacts. In fact, datasets such as SWAG and RECLOR evaluated in this paper were produced using variants of this filter approach which iteratively perturb the data instances until a target model can no longer fit the resulting dataset well. Some methods (Yaghoobzadeh et al., 2019), instead of preprocessing the data by removing biases, leave out samples with biases in the middle of training according to the decision made between epoch to epoch. (Bras et al., 2020) investigated model-based reduction of dataset cues and designed an algorithm using iterative training. Any model can be used in this framework. Although such an approach is more general and more efficient than human annotating, it heavily depends on the models. Unfortunately, different models may catch different cues. Thus, such methods may not be complete.

# 7 Conclusion

Our lightweight framework, ICQ, identifies biases and cues in multiple-choice NLU datasets, illuminating model behaviors from a statistical perspective. Extensive experimentation on diverse tasks validates ICQ's efficiency in revealing dataset and model biases. Using a case study on ChatGPT, we explore its cues, offering practical guidance. ICQ advances our understanding and optimization of large language models, promoting the creation of robust, unbiased AI systems.

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
