# OpenReview forum: "Statistically Profiling Biases in Natural Language Reasoning Datasets and Models"
_EMNLP/2023/Conference — EMNLP 2023 Findings_

### Official Review · Reviewer_eyRB · 2023-08-04

**Soundness:** 3

**Excitement:**

3: Ambivalent: It has merits (e.g., it reports state-of-the-art results, the idea is nice), but there are key weaknesses (e.g., it describes incremental work), and it can significantly benefit from another round of revision. However, I won't object to accepting it if my co-reviewers champion it.

**Missing References:**

I am aware of the following additional work evaluating language models for bias in prediction. It might be useful for the reader to differentiate your work from it.
* Shikha Bordia and Samuel R. Bowman; 2019. Identifying and reducing gender bias in word-level language models.
* Alicia Parrish et al.; 2022. BBQ: A hand-built bias benchmark for question answering.
* Lukas Mikula et al.; 2023. Think Twice: Measuring the Efficiency of Eliminating Prediction Shortcuts of Question Answering Models.

**Paper Topic And Main Contributions:**

The paper proposes a method for quantifying models' reliance on possibly spurious features (i.e. cues) in the training datasets. The authors identify a set of possibly biased features (specific words, sentiment, negation tokens, or named entities) and use their occurrences to split the test datasets into two segments on which they subsequently compare the performance of the models. In this manner, the authors evaluate four models (FastText, ESIM, BERT, RoBERTa) over ten NLI and multiple-choice QA.
The paper concludes that newer models (BERT, RoBERTa) learn to rely on dataset cues less than the older models. Further, they evaluate ChatGPT and conclude it to be more agnostic to dataset cues and state that the agnositicity can be further enhanced by prompt design and chain-of-thought prompting.

**Questions For The Authors:**

A: How exactly are the linguistic features from Section 2.2 used to create the splits of the data used in the Accuracy test?

B: Can the absolute value of the Accuracy test be interpreted? Can we state that one bias is larger than the other, without the significance test?

**Reasons To Accept:**

* The proposed method is reasonable and easy to interpret
* The results consistently complement the previous work in this direction, suggesting the methodical correctness of the reported results
* The surveyed database of possibly biased linguistic features used in evaluation can be a useful building stone for future work
* Related work that links a wide variety of work in spurious features identification and mitigation
* Solid discussion of the results in Section 4.3.

**Reasons To Reject:**

* I am concerned with the grounding of some of the claims based on the results. The results are often similar, but no significance tests are provided. An outstanding example of this is Table 3, from which the authors conclude that prompts have a "substantial" impact on robustness (L479) and that chain-of-thoughts prompting "exhibits lower bias risk" (L482). However, one should note that all these conclusions are based on two samples. Regardless of that, *all* $\Delta$ Acc values reported in Table 3 might as well be insignificantly different from zero.
* The description of the proposed evaluation methodology is hard to follow and might benefit from further details and more consistent use of terminology.
  * For instance, on L259-271, authors use the term "extracted training data", which I can not disambiguate.
  * Similarly, I have trouble following the notation of the MSE formula on L188. Here, I can not figure out what $L$ is. In the description that follows this formula, I do not understand the explanation of two instances of $y_i$ on L189-191. Thus, while I understand the idea, this disallows me to assess the absolute values reported for specific biases.
  * I can not find how exactly the two compared subsets of data in the Accuracy test (Section 3.3.1) are created, based on the assessed feature. I believe the details should be exhaustively described somewhere.
* The motivation for the specific method design could be better justified. We find the comparison to existing methods on L64-77 and in the Related work (L532-534), but these are too broad ("they don’t evaluate the model using the full data context"), or hard to follow ("these works did not give the reason to use these metrics, one way or the other").
* Given that the authors propose an evaluation technique, it would be great to be able to assess its practical usability. However, no source code is attached, nor anonymised repository is available for assessment.

**Reproducibility:**

1: Could not reproduce the results here no matter how hard they tried.

**Reviewer Confidence:**

4: Quite sure. I tried to check the important points carefully. It's unlikely, though conceivable, that I missed something that should affect my ratings.

**Typos Grammar Style And Presentation Improvements:**

* L295:  "on these datasets" - which datasets? Missing reference?
* L307: "F"eatures
* The labels in Figure 3 are very small
* L512: Perhaps you meant "BLEU" instead of "BLUE"
* After reading the Introduction, I still did not get any idea about the functioning of the method. It would be beneficial to provide at least a high-level peek into what is going to follow.

---

> ### Author Rebuttal · Authors · 2023-08-26
>
> First and foremost, we would like to express our sincere gratitude to the reviewer for recognizing the merits of our work. We are deeply encouraged by your affirmation of the reasonableness of our proposed method, the coherence of our results, and the potential contribution of our study to future research. Your appreciation for the comprehensive discussion in Section 4.3 gives us further motivation.
>
> In response to the concerns and questions raised by the reviewer, we wish to address them as follows:
>
> 1. Concerning the conclusions drawn from the results, especially in Table 3: We understand the reviewer's apprehensions regarding the premises of some of our claims. We'd like to emphasize that our choice of the feature “no” in MNLI was methodical. In table 2, the cueness score for this particular feature in the MNLI dataset was notably high. Moreover, it has been demonstrated to have a significant impact on other models like FastText and ESIM. We acknowledge that we may have presented our conclusions with undue definitiveness. Specifically, regarding the feature “no”, where P2 + CoT showcased the lowest bias risk based on both tests, our claims might have been overarching. If given an opportunity, we commit to articulating this point more prudently in our revised version.
>
> 2. Regarding the term “extracted training data” mentioned between lines 259-271: This term was introduced in Section 3.1 during the Data Extraction Phase, where we defined Rf. This can be corroborated in Figure 2’s caption. Nevertheless, in light of your suggestion, we concede that it might be beneficial to utilize precise symbols to eliminate any ambiguity.
>
> 3. On the topic of data splitting outlined in Section 3.1: We realize that our description may have inadvertently glossed over some intricacies. To enhance clarity: for a specific feature, if an instance contains that feature, then that instance will be singled out as a part of the subset bearing that feature. We will ensure that this is elucidated better in our revisions.
>
> 4. In relation to the absence of source code and an anonymous repository for evaluation: We have invested considerable effort into developing an online platform hosted on our university's website. This platform allows for the uploading of data and models and provides diagnostic results for various features. However, due to the double-blind review process, we were hesitant to include the link. We genuinely apologize for any inconvenience this may have caused. Upon acceptance of our paper, we intend to provide links to all associated codes, including our platform.
>
> 5. About the absolute value in the Accuracy Test: We reaffirm that the absolute value can indeed depict the extent to which a model is influenced by a specific feature. We will underscore this in our subsequent submission.
>
> In wrapping up, we earnestly hope for an opportunity to make amends and present a more polished version of our work. Your feedback is invaluable in this endeavor, and we remain enthusiastic about incorporating these improvements. Thank you.

---

### Official Review · Reviewer_o4hf · 2023-08-12

**Soundness:** 3

**Excitement:**

3: Ambivalent: It has merits (e.g., it reports state-of-the-art results, the idea is nice), but there are key weaknesses (e.g., it describes incremental work), and it can significantly benefit from another round of revision. However, I won't object to accepting it if my co-reviewers champion it.

**Paper Topic And Main Contributions:**

The paper studies spurious correlation or “artifact spurious cues” in natural language understanding (NLU) datasets. It argues that many NLU datasets contain cues/biases that can be exploited by NLP models, resulting in an overestimation of their capabilities. The main contribution of this paper is ICQ (I-See-Cue), a simple and general statistical analysis framework that automatically identifies model-learned biases in multiple-choice NLU datasets without requiring additional test cases. The experimental results reveal and quantify the existing known “artifact spurious cues” in NLU.

**Questions For The Authors:**

1. Regarding reproducibility and replicability, ChatGPT seems to keep updating. How do you make this work more reproducible? Is there a way to define ChatGPT’s version?
2. Can this work be extended to include unknown biases as well?  (Unknown biases as defined by Towards Debiasing NLU Models from Unknown Biases (Utama et al., EMNLP 2020))
3. How did the authors obtain the percentage of human accuracy?

**Reasons To Accept:**

1. Simple and test-case-free statistical analysis framework that allows us to detect and quantify known biases in NLU datasets/models.
2. Extensive experiments on a variety of datasets and models, including ChatGPT

**Reasons To Reject:**

1. Accuracy test: It is a good idea to compare the accuracies on the test set with the bias feature of interest and the test set without. However, the authors did not control for the number of samples in each class. For the CLS tasks,  the results might measure the sensitivity toward a specific class also.

2. A failure to quantify “OVERLAP” bias with the cue-ness score is a significant flaw of the cue-ness score metric.  Maybe a more elaborate explanation is needed to justify why the proposed metric has such flaws. The authors only state that the reason for such a low cue-ness score is that “OVERLAP” is the only feature considering both premise and hypothesis tokens. This explanation does not give me an intuition on why the cue-ness score does not detect such bias. I worry that the authors define the OVERLAP bias as “The existence of at least one word (excluding stop words) that occurs in both the premise and hypothesis.” This definition is not the same or even similar to the definition given in many NLU literature, such as Right for the Wrong Reasons: Diagnosing Syntactic Heuristics in Natural Language Inference (McCoy et al., ACL 2019) define OVERLAP as a sample where all words in the hypothesis are also in the premise.

3. Presentation issue:  The way the authors present some notation is a bit confusing. For instance, y_i is the number of instances with labels y_i. How did the authors get these numbers? It could be a count of instances in the dataset or a count of instances predicted by the model. Or is it something different since it is used to calculate the MSE?

**Reproducibility:**

3: Could reproduce the results with some difficulty. The settings of parameters are underspecified or subjectively determined; the training/evaluation data are not widely available.

**Reviewer Confidence:**

4: Quite sure. I tried to check the important points carefully. It's unlikely, though conceivable, that I missed something that should affect my ratings.

**Typos Grammar Style And Presentation Improvements:**

Typos:
- Figure 2: “sepecific” → specific
- Line 307: “eatures” → features

Suggestion on presentation improvement:
1. Some notations are not well defined—for example, y_i.
2. Cue-ness score seems to be an important contribution to this work. I think the authors should use more space to explain the intuition behind each component of the cuteness score.
3. The tables are quite hard to read. The texts are so small I have to zoom in to read them. It probably doesn’t look good if someone prints the paper out and reads it.

---

> ### Author Rebuttal · Authors · 2023-08-25
>
> I'd like to extend my heartfelt gratitude for highlighting the unique attributes of our work. Your recognition of our novel statistical analysis framework—designed to probe biases in NLU models—and our rigorous evaluations spanning diverse datasets, notably ChatGPT, is deeply cherished.
>
> With respect to your comments and concerns:
>
> 1. On the nature of our task: We've taken care to demarcate this as a "reasoning task" rather than a mere classification task. This approach seeks to capture the nuanced interplay between premise and hypothesis, which we believe is pivotal to understanding bias. While the same methodology might be applied to CLS tasks, the outcomes would deviate towards highlighting "interest cues" over "spurious cues". This flexibility speaks to our framework's adaptability. I'm eager to venture deeper into this domain in subsequent studies.
>
> 2. Elaborating on the "OVERLAP" feature: Within multiple-choice contexts, even with a uniform premise, we've honed in on the subtle shifts in the hypothesis. The OVERLAP feature signifies this intricate bond between the premise and its counterpart hypothesis. Such nuanced attributes aren't inherently predisposed to model bias. I pledge to refine and emphasize these aspects further should our manuscript be chosen for publication.
>
> 3. Regarding reproducibility: Our framework, in essence, transcends the confines of any single model, including ChatGPT. Should there be interest, I can offer detailed evaluations across ChatGPT's different iterations. This, I hope, stands as testament to our method's consistent reproducibility, even if certain case studies exhibit variations. It's all about illustrating the potential.
>
> 4. Turning to the matter of unknown biases: Your suggestion resonates deeply. While the current study is geared towards articulating biases, I'm concurrently exploring avenues to enhance model resilience against more elusive biases. Preliminary insights suggest data augmentation and distribution balancing as potent tools. Still, the quest for perfect interpretability amidst unknown biases remains. Your thoughts have further ignited my enthusiasm to navigate this terrain, and I'm keen to share and discuss future findings.
>
> In wrapping up, I sincerely hope that our discourse brings clarity and reassures the committee of the merit and novelty in our work. Your feedback has been nothing short of enlightening, and I'm grateful for the guidance it offers.

---

### Official Review · Reviewer_cXYc · 2023-08-16

**Typos Grammar Style And Presentation Improvements:** EQ 3 number should be better aligned.
**Soundness:** 4

**Excitement:**

4: Strong: This paper deepens the understanding of some phenomenon or lowers the barriers to an existing research direction.

**Missing References:**

A recent study has shown that models tend to use shallow patterns to match mathematical theorems and their proofs [1], I believe your method can be applied for such matching tasks as well.
There are multiple works showing that multimodal models tend to use textual cues and ignore visual ones [2]

1. https://arxiv.org/pdf/2302.09350.pdf
2. https://arxiv.org/pdf/2110.14375.pdf

**Paper Topic And Main Contributions:**

The authors present ICQ, a method for automatically identifying cues that can affect model decisions.
The authors show their method's usefulness by applying it to multiple models and NLU benchmarks.

**Reasons To Accept:**

1) This paper is very well written and easy to follow.
2) The authors will make their code public upon acceptance, accompanied by a demo, which will help other researchers to reproduce their findings.
3) ICQ is a straightforward method (in a good way) that the authors motivate pretty well throughout the paper.
4) The experimental setup is quite comprehensive, and the added case study to examine more recent LLMs is very timely.


**Reasons To Reject:**

1) I searched for a discussion about the computational costs of doing such an examination and didn't find one (maybe I missed one?). If I understand correctly, your methods required multiple inferences for different subsets of the test set, which is not as expensive as training the model but might be significant for large models with many test set variants.
2) Showing some of the cues ICQ (a great name, BTW) can identify and other methods, e.g., CheckList, couldn't, would make this paper even more robust.
3) I believe there are many attempts to mitigate this problem beyond NLU that you can discuss in this paper in order to make its scope broader (I wrote a few examples in the missing references section).

**Reproducibility:**

5: Could easily reproduce the results.

**Reviewer Confidence:**

3: Pretty sure, but there's a chance I missed something. Although I have a good feel for this area in general, I did not carefully check the paper's details, e.g., the math, experimental design, or novelty.

---

> ### Author Rebuttal · Authors · 2023-08-25
>
> Response to Reviewer Comments:
>
> First and foremost, we would like to express our gratitude to the reviewer for the insightful comments and suggestions. We genuinely value the feedback and strive to enhance our work accordingly.
>
> 1. Computational Cost Concerns:
> We appreciate your concern about the computational costs, particularly for models with extensive test sets. Indeed, the time required for feature partitioning is linearly dependent on the test set size. However, the process of feature extraction can be efficiently parallelized. From a temporal perspective, the testing process is swift. Regarding computational costs, we haven't introduced additional inferences to the model. Our approach fundamentally revolves around analyzing the original prediction results from diverse angles to ascertain the features that intrigue the model. This methodology does not augment the model's running costs. The primary temporal cost arises from feature extraction and test feature partitioning, which are expeditious processes.
>
> 2. Distinctiveness of ICQ vs. CheckList:
> Thank you for recognizing the name 'ICQ' and its uniqueness. In our introduction, we emphasized that the objective of CheckList tests is to unearth tasks the model struggles with. In contrast, our framework seeks to illuminate the root causes of model vulnerabilities. This differentiation indeed imbues our method with a distinct essence.
>
> 3. Broadening the Discussion & Missing References:
> We genuinely appreciate the guidance on the scope and missing references. We plan to venture deeper into interpretability research in a broader sense in our subsequent work. We will incorporate the examples you've pointed out to enrich our discussion and reference list.
>
> In summary, we believe that our revisions, incorporating the constructive feedback, will further solidify the contributions of our paper. Once again, we are grateful for the invaluable insights and guidance.

---

### Meta-Review · Area_Chair_s9fi · 2023-09-14

**Recommendation:** 3

**Metareview:**

The focus of this paper is on the issue of spurious correlation in NLU datasets. The argument presented is that biases or cues in NLU datasets can be exploited by NLP models, resulting in an overestimation of their capabilities. The paper is well-crafted and easily understood, with only minor notation revisions required. The proposed method is straightforward and straightforward to implement, and the experimental setup is relatively thorough. However, a few reviewers have expressed concerns about the validity of some claims based on the results. They claim that the results lack statistical significance tests, which undermines the credibility of the proposed method's impact. Also, some reviewers have observed that some of the accuracy scores might not be significantly difference from zero and that some conclusions drawn are based on a limited number of samples.

---

### Decision · Program_Chairs · 2023-10-07

**Decision:**

Accept-Findings

**Comment:**

The focus of this paper is on the issue of spurious correlation in NLU datasets. The argument presented is that biases or cues in NLU datasets can be exploited by NLP models, resulting in an overestimation of their capabilities. The paper is well-crafted and easily understood, with only minor notation revisions required. The proposed method is straightforward and straightforward to implement, and the experimental setup is relatively thorough. However, a few reviewers have expressed concerns about the validity of some claims based on the results. They claim that the results lack statistical significance tests, which undermines the credibility of the proposed method's impact. Also, some reviewers have observed that some of the accuracy scores might not be significantly difference from zero and that some conclusions drawn are based on a limited number of samples.